# A geospatial analysis comparing wastewater-monitored sewershed and statewide populations for 32 states participating in CDC's National Wastewater Surveillance System, 2021-2024

Stacie Reckling[1,2]*, Heather Reese[1], Lisa P. Oakley[1], Helena Mitasova[2,3]

**1** Centers for Disease Control and Prevention, Atlanta, Georgia, United States of America, **2** Center for Geospatial Analytics, North Carolina State University, Raleigh, North Carolina, United States of America, **3** Marine Earth and Atmospheric Sciences, North Carolina State University, Raleigh, North Carolina, United States of America

* skreckli@ncsu.edu

## Abstract

Wastewater monitoring analyzes samples of untreated wastewater for pathogens to track community-level disease trends. A sewershed polygon helps define the sample population by depicting the community area that contributes sewage to the wastewater sample. This study utilized sewershed geospatial data submitted to the Centers for Disease Control and Prevention's National Wastewater Surveillance System (CDC NWSS) to determine the wastewater-monitored population characteristics and assess how well the wastewater-monitored population represents the broader population. In a geographic information system, we intersected sewershed, state, and county polygons with US Census data and social vulnerability data to calculate the proportion of the population with certain demographic and social vulnerability characteristics in each geographic unit. In 32 states, we compared the aggregated sewershed population within the state to the statewide population for wastewater monitoring sites sampling during the Fall of 2024. In four states, Colorado, New York, North Carolina, and Wisconsin, we further assessed wastewater representativeness; we compared the distributions of sewershed populations with county populations and examined how representativeness has changed across eight time points during 2021–2024. In Fall of 2024, CDC NWSS included sites in all 50 states covering 41% of the US population. In 5 of 32 states analyzed the wastewater-monitored population represented the statewide population across all variables except median household income. In 27 states, we observed that wastewater-monitored populations may over- or underrepresent certain populations based on factors like race and ethnicity, education, or social vulnerability. We found that over time, differences between sewershed and statewide population characteristics were only slightly affected by changes in the number of sampling sites and population. State and local health departments, as well as CDC NWSS, can utilize information about the wastewater sample population and

**Data availability statement:** Links to publicly available databases used in this project are provided in the reference section. The data supporting the findings of this study are available in Supplemental Information. Sewershed polygons are unavailable due to wastewater utilities' ethical concerns about disclosing protected infrastructure information and can be requested directly from the wastewater utilities.

**Funding:** The authors received no specific funding for this work.

**Competing interests:** The authors have declared that no competing interests exist.

how it compares to the state population to strengthen the use of wastewater monitoring data for public health action.

## Introduction

Approximately 84% of the United States' population is connected to a public sewer system [1]. However, sewer connectivity is not equally distributed in the US; western regions and urban areas tend to have higher rates of sewer connectivity [2]. Wastewater monitoring, a crucial public health tool, utilizes existing wastewater infrastructure to collect a sample of untreated wastewater which is analyzed for human pathogens. Initially in response to the COVID-19 pandemic, wastewater monitoring programs were launched nationwide to track disease trends in the population within a sewershed, or the community area upstream of a wastewater sampling site [3]. These data provide a community-level estimate of disease that is not impacted by an individual's access to healthcare or symptom severity. Throughout the pandemic, wastewater data served as an early indicator of increasing trends and became increasingly relied upon as at-home testing became commonplace and case-based surveillance became less reliable [4,5]. Wastewater monitoring in the US has since expanded to include influenza viruses, including influenza A subtypes during an outbreak of avian influenza A (H5), respiratory syncytial virus (RSV), and other outbreak pathogens such as monkeypox virus, and measles virus [6,7]. Public health officials analyze wastewater data alongside clinical surveillance data to gain a more comprehensive understanding of the diseases present in the community.

Wastewater represents a composite sample of the community upstream of a collection site. Although metadata submitted to the Centers for Disease Control and Prevention's National Wastewater Surveillance System (CDC NWSS) does not include detailed demographic information, the accompanying geographic metadata can be used to infer characteristics of the wastewater-monitored population. CDC NWSS collects different types of sample site location data including the county(ies) served and a sewershed polygon representing the geographic area that contributes wastewater to that sampling site. In 2022, when sewershed data submitted to NWSS were still largely incomplete, NWSS populations sampled were described in terms of the overall social vulnerability and urbanicity of the county(ies) served by the sampling sites [3]. They observed that counties with NWSS testing were more urban and had slightly lower overall social vulnerability as compared to all US counties. A detailed evaluation of North Carolina's sewershed populations found that individual sewershed populations sometimes differed from county populations but that characteristics of wastewater-monitored sewershed populations aggregated to the state level were similar to statewide population characteristics [8].

According to the National Academies of Science Engineering and Medicine, an important characteristic of a national wastewater surveillance system is equitable distribution among population demographics [9,10]. A better understanding of the

sewershed community within wastewater monitoring networks can help to meet this objective. Sewershed population information also provides valuable context to wastewater data which will improve its interpretation and actionability. However, a lack of a national sewershed dataset has prevented the completion of a comprehensive geospatial analysis characterizing wastewater-monitored population representativeness. We leveraged sewershed geospatial data submitted to NWSS to investigate the wastewater-monitored population through two research questions: 1) Do sewershed populations monitored through CDC NWSS differ from the statewide populations with respect to certain demographic and social vulnerability characteristics, and if so, how do they differ and 2) have the differences changed as sampling sites were enrolled or dropped over time? This analysis highlights similarities and differences between monitored sewershed populations and broader populations and increases our understanding of how wastewater data can be used to support public health and public health action.

## Methods

### Data collection and pre-processing

Wastewater concentration data were submitted to CDC NWSS by state, tribal, local, and territorial (STLT) health departments, a CDC-managed contract for commercial testing, intended to supplement the geographic coverage of the STLT programs, and a private-academic partnership program. Sewershed geospatial data were submitted to CDC NWSS by STLT programs for their sites as well as commercial testing sites in their state. Based on the wastewater data received by NWSS from STLT and commercial testing sites as of November 2024, we generated a list of current sites, or those that had submitted sampling results during the most recent 3 months (August 2024 – October 2024). To investigate population changes over time, we created lists of sites at eight different time points (January and August of the years 2021, 2022, 2023, and 2024) to reflect sites sampling during peak and off-peak respiratory virus season. For each list of sampling sites, we joined the tabular data to georeferenced sewershed polygons, keeping the sewersheds that matched with a sampling site on the list. State boundary geospatial data was downloaded from the 2020 US Census [11]. A state was included in the analysis if greater than 80 percent of their Epidemiology and Lab Capacity (ELC) funded wastewater treatment plant (WWTP) sampling sites or greater than 80 percent of their sampled population had a corresponding sewershed polygon. This activity was reviewed by CDC and was conducted consistent with applicable federal law and CDC policy.

Variables describing race, ethnicity, and group quarters (populations living or staying in a group living arrangement often with specific services or care provided) were obtained from the 2020 United States Census Redistricting Data, which were available at the census block level (S1 Table) [12]. We also analyzed variables from the 2018–2022 American Community Survey (ACS) that captured information about age, health insurance status, educational level, wealth, English proficiency, housing, unemployment, and disability status, all of which were available at the census tract level (S2 Table). American Community Survey 2018–2022 data at the tract level were obtained from ESRI's Living Atlas and compiled in ArcGIS Pro [13]. Sewershed total population estimates were derived from the Environmental Protection Agency's EnviroAtlas Dasymetric Allocation of Population, 2020 raster data which allocates population counts from Census data to inhabitable areas at a 30-meter resolution, making it particularly suitable for irregularly shaped sewersheds that cross multiple administrative boundaries [14]. Lastly, we used a shapefile of nationwide Census tracts with information on the CDC/ATSDR Social Vulnerability Index 2022 (SVI) [15]. SVI analyzes variables from US census data to assess and rank 16 social factors such as poverty, crowded housing, race, and disability, to determine the social vulnerability of each Census tract or county across the United States. In addition to the social vulnerability percentile ranks for overall social vulnerability, the index also includes vulnerability related to four themes, including socioeconomic status (SES), household characteristics, racial and ethnic minority status, and housing type and transportation (S1 Table). We analyzed overall SVI, theme-level SVI, and individual Census and ACS variables to understand whether populations differed with respect to a particular type of social vulnerability, or overall, and whether a given variable was driving vulnerability.

## Geospatial analysis

We calculated the total population, the proportion of the population in each Census variable, and the average social vulnerability ranking for each polygon dataset of interest which included states, counties, the state's aggregated sewersheds, and individual sewersheds. To produce a multi-part polygon representing the state's aggregated sewershed, we merged the individual monitored sewershed polygons by state. This allowed us to analyze the statewide sewershed population without double-counting populations when sewersheds shared a boundary. We calculated the sewershed total population by overlaying sewersheds with the EnviroAtlas dasymetric population raster. To analyze the 2020 US Census and ACS demographic variables, we selected Census blocks or tracts that intersected each polygon of interest and weighted the total population according to the area of the block or tract that was inside the sewershed polygon. Then, the selected tracts or blocks were merged and summary statistics were calculated including the sum of the population and the average median household income. Population counts were converted to percentages by dividing by the total population using total population in blocks or tracts where appropriate. We also calculated the population-weighted social vulnerability ranking of populations based on SVI 2022 tract level data for overall SVI and SVI related to the four themes. All geospatial analyses were conducted using Python (pandas, geopandas, rasterio, rasterstats, and scipy packages) [16–20].

To compare the aggregated sewershed population and the statewide population, we calculated the percentage point (%) difference for categorical variables and the percent difference for continuous variables. We assessed the sensitivity of various difference thresholds (plus or minus 1%, 3%, 5%, 7%, and 10%) to identify the threshold that could capture potentially meaningful differences. We chose +/-5% because across most variables, it balanced the need to capture subtle yet real distinctions while minimizing noise from minor variations and aligned with previous similar analyses [8]. In addition to the absolute difference, we calculated the ratio of the sewershed population proportion to the statewide population proportion to highlight differences for variables with small populations such as the proportion living in group quarters or unemployed. We did not test for statistically significant differences due to the non-normal distribution and nested structure of the data (the monitored sewershed population were a subset of the state population).

In a sub-analysis of the first research question, we assessed representativeness at a finer geographic scale to explore whether the characteristics of highly populated sewersheds were masking the characteristics of less populated sewersheds when sewersheds aggregated to the state level were compared to the state. We analyzed demographic and social vulnerability of populations in individual sewersheds and counties in a subset of states, Colorado, New York, North Carolina, and Wisconsin, which had complete sewershed polygon data and represented different regions of the US. As described above, we converted population counts to population proportions for each monitored sewershed (sampling August - October 2024) and each county in the state. Then, a two-sample Kolmogorov-Smirnov test (KS-test) was performed for each variable to compare the distribution of the values between sewersheds and counties within each state. The KS-test, which is sensitive to differences in both the center and the shape of the distributions, produced a KS statistic, indicative of the maximum difference between the distributions, and an associated p-value (p-value < 0.05 were significant). Kernel density estimate (KDE) plots were used to visually assess the direction of differences between the sewershed population distribution relative to the county population distribution.

To learn whether representativeness has changed as sampling sites were added or removed over time, we compared the aggregated sewershed population and the statewide population at eight time points (January and August, 2021–2024). We did this for a subset of states, New York, Wisconsin, Colorado, and North Carolina, which were early adopters of wastewater monitoring, had complete sewershed polygon data for sites sampling between 2021 and 2024, and represented different regions of the US. Using the geospatial data and methods already described, we analyzed the demographics and SVI of the state's aggregated sewershed population at each time point. We then compared the sewershed population to the state population by calculating the difference between the proportion of the sewershed population and the state population for each variable.

## Results

Thirty-two states met the site or population coverage criteria and were included in the analyses (Table 1). The number of monitored sewersheds per state was highest in New York and lowest in Vermont (188 and 2, respectively), with a median of 21 monitored sewersheds per state. Utah included the highest percentage of the statewide population in monitored

**Table 1. Wastewater-monitoring coverage in 32 states analyzed.**

| State name | State population (2020) [a] | Monitored sewershed population (2020) [b] | % State population in monitored sewershed | # Monitored counties | # Monitored sewersheds | # Monitored sites |
|---|---|---|---|---|---|---|
| Arizona | 71,51,502 | 30,02,238 | 41.98 | 15 | 14 | 15 |
| California | 3,95,38,223 | 1,64,10,707 | 41.51 | 58 | 49 | 54 |
| Colorado | 57,73,714 | 31,15,113 | 53.95 | 64 | 20 | 20 |
| Delaware | 9,89,948 | 6,28,020 | 63.44 | 3 | 6 | 12 |
| Georgia | 1,07,11,908 | 19,18,949 | 17.91 | 159 | 14 | 18 |
| Hawaii | 14,55,271 | 9,71,777 | 66.78 | 5 | 9 | 9 |
| Indiana | 67,85,528 | 21,82,469 | 32.16 | 92 | 26 | 26 |
| Kansas | 29,37,880 | 5,38,353 | 18.32 | 105 | 9 | 9 |
| Kentucky | 45,05,836 | 15,48,348 | 34.36 | 120 | 22 | 22 |
| Maine | 13,62,359 | 2,02,499 | 14.86 | 16 | 19 | 21 |
| Maryland | 61,77,224 | 4,50,056 | 7.29 | 23 | 9 | 9 |
| Minnesota | 57,06,494 | 31,47,757 | 55.16 | 87 | 27 | 35 |
| Missouri | 61,54,913 | 30,21,042 | 49.08 | 114 | 62 | 64 |
| Nebraska | 19,61,504 | 13,00,156 | 66.28 | 93 | 16 | 16 |
| New Hampshire | 13,77,529 | 2,66,735 | 19.36 | 10 | 14 | 14 |
| New Jersey | 92,88,994 | 46,05,440 | 49.58 | 21 | 21 | 22 |
| New Mexico | 21,17,522 | 9,29,094 | 43.88 | 33 | 9 | 11 |
| New York | 2,02,01,249 | 1,55,57,790 | 77.01 | 62 | 188 | 189 |
| North Carolina | 1,04,39,388 | 29,00,139 | 27.78 | 100 | 35 | 36 |
| Ohio | 1,17,99,448 | 56,26,205 | 47.68 | 88 | 75 | 75 |
| Oklahoma | 39,59,353 | 15,88,676 | 40.12 | 77 | 18 | 18 |
| Oregon | 42,37,256 | 24,15,094 | 57.00 | 36 | 31 | 37 |
| Pennsylvania | 1,30,02,700 | 20,61,718 | 15.86 | 67 | 33 | 38 |
| Rhode Island | 10,97,379 | 5,67,678 | 51.73 | 5 | 11 | 12 |
| South Carolina | 51,18,425 | 12,24,836 | 23.93 | 46 | 23 | 24 |
| Utah | 32,71,616 | 28,24,508 | 86.33 | 29 | 35 | 35 |
| Vermont | 6,43,077 | 8,646 | 1.34 | 14 | 2 | 2 |
| Virginia | 86,31,393 | 43,97,442 | 50.95 | 95 | 31 | 38 |
| Washington | 77,05,281 | 40,26,997 | 52.26 | 39 | 30 | 30 |
| West Virginia | 17,93,716 | 3,18,544 | 17.80 | 55 | 24 | 27 |
| Wisconsin | 58,93,718 | 27,70,710 | 47.01 | 72 | 42 | 43 |
| Wyoming | 5,76,851 | 1,69,027 | 29.30 | 23 | 7 | 7 |

A table showing information about the percentage of the state population included in wastewater monitoring, counties with wastewater monitoring, monitored sewersheds, and monitored wastewater sites sampling (August to October 2024) for 32 states analyzed. A state was included in the analysis if it had a sewershed for at least 80% of its monitoring sites or monitored population. Sewershed and state populations were estimated from US Census 2020 data to ensure that the percentage of the statewide population included in wastewater monitoring was calculated from the same year's data.

[a]State populations were obtained from the 2020 US Census data.

[b]Sewershed populations were estimated from the EPA Dasymetric Allocation of Population US, 2020.

sewersheds (86%) while Vermont had the lowest statewide population coverage (1%). The median percentage of the state population included in wastewater monitored sewersheds was 42%.

## Comparing characteristics of aggregated sewershed and statewide populations

To address our first question, we compared the aggregated sewershed population within a state to the statewide population for 26 variables obtained from US Census, ACS, and SVI data. Across all states, the current monitored sewershed and statewide populations were similar (less than +/-5% difference) for 11/26 variables (less than 5 years, 65 years or older, no health insurance, disability, below federal poverty level, unemployed, native Hawaiian/Pacific Islander, limited English speaking, and group quarters - correctional, nursing, and military; Fig 1). Across 31 states there were four additional variables that had a less than +/-5% difference (less than 18 years, limited English speaking, Asian, and American Indian and Alaska Native; Fig 1). We found meaningful differences (greater than +/-5%) between the sewershed and state populations in multiple states for 7/26 variables (White alone, African American alone, Hispanic, median household income, bachelor's degree or higher, racial and ethnic minority status SVI, and housing type and transportation SVI; Fig 1). In general, the monitored sewershed populations had a lower proportion of Whites and a higher proportion of either African American, Hispanic, or Asian persons compared to the statewide population (Fig 1). The ratio between the sewershed and statewide population further demonstrated that Asians were over-represented in wastewater and that sewershed populations included a higher proportion of limited English speakers. Also, monitored sewershed populations tended to have higher educational attainment (Fig 2c) and lower median household income (Fig 2d). The ratio between income in the sewershed and income in the state better exemplified the income gap; the sewershed populations' income was 25% lower in half of states analyzed and 50% lower in 5 states. Differences in individual race and ethnicity characteristics were often reflected as meaningfully higher racial and ethnic minority vulnerability in the monitored sewershed population (Fig 1 and 2b). We also observed that overall social vulnerability and vulnerability related to each theme were often meaningfully different between the sewershed and the state population, and that the direction of the difference (higher or lower in the sewershed) often varied across states (Fig 1 and 2a). Finally, based on the population ratio, we also saw differences in the proportion of the population in group quarters where, generally, sewersheds included a lower percentage of people living in correctional group quarters and a higher percentage of people living in college group quarters.

However, in some states, the difference between the monitored sewershed and statewide population characteristics deviated from the patterns observed in most states. In Maryland, the monitored sewershed population had a meaningfully higher proportion of Whites, a lower proportion of minorities, and lower social vulnerability overall and across all themes which was opposite to what was observed in other states (Fig 1). The pattern was also different in Kansas, where we found that the sewershed and statewide populations were similar in race and ethnicity and that the sewershed population had meaningfully lower social vulnerability rankings (except for racial and ethnic minority status vulnerability) compared to the statewide population. A higher proportion of people living in group quarters, specifically colleges, was identified only in Vermont and New Hampshire's monitored sewershed populations (Fig 1). Finally, some states' wastewater monitored populations represented the statewide population well for measured variables. We observed that five states, Colorado, Nebraska, Oregon, South Carolina, and Utah, did not have any variables where the monitored sewershed and the statewide populations differed by greater than +/-5% (excluding median household income, which was meaningfully lower in the sewershed across all states; Fig 1). A complete list of differences between the sewershed and statewide population for each state and variable analyzed can be found in the supplemental material (S2 Table).

## Comparing population characteristics across individual sewersheds and counties

In the sub-analysis, the relationship between the distributions of populations in monitored sewersheds and populations in counties differed across the four states analyzed. In North Carolina, a comparison of populations in 35 monitored sewersheds with populations in the state's 100 counties showed that the sewershed distribution had significantly different

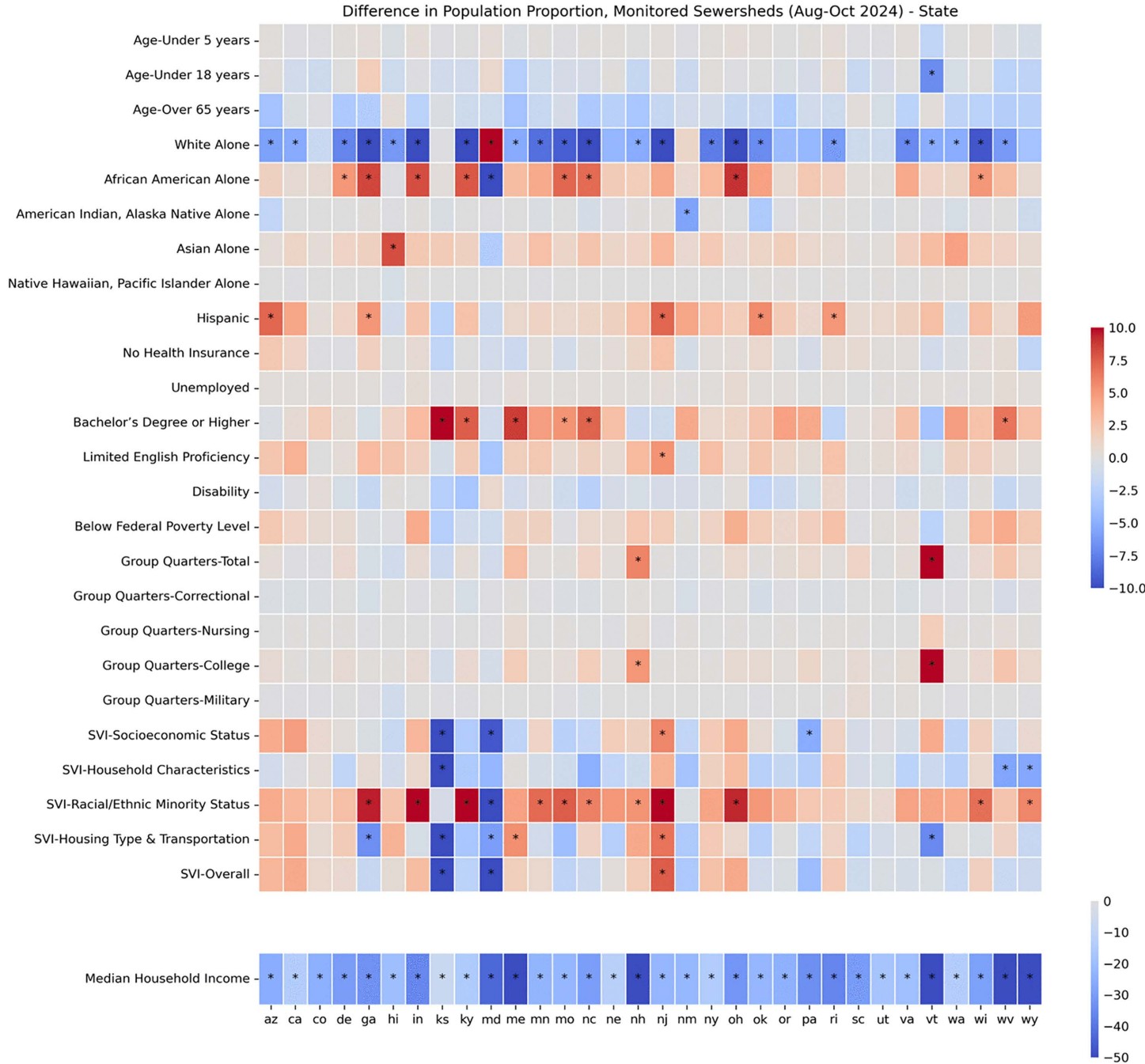

**Fig 1. Differences between statewide monitored sewershed populations and the statewide populations.** A heatmap showing the percentage differences between the aggregated statewide monitored sewershed population proportion and the statewide population proportion for sites sampling August to October 2024. Shades of red indicate the proportion of the population is higher in the sewershed and shades of blue indicate the proportion of the population is lower in the sewershed. An * indicates that the difference is greater than +/- 5%. State abbreviations are displayed along the x-axis in alphabetical order.

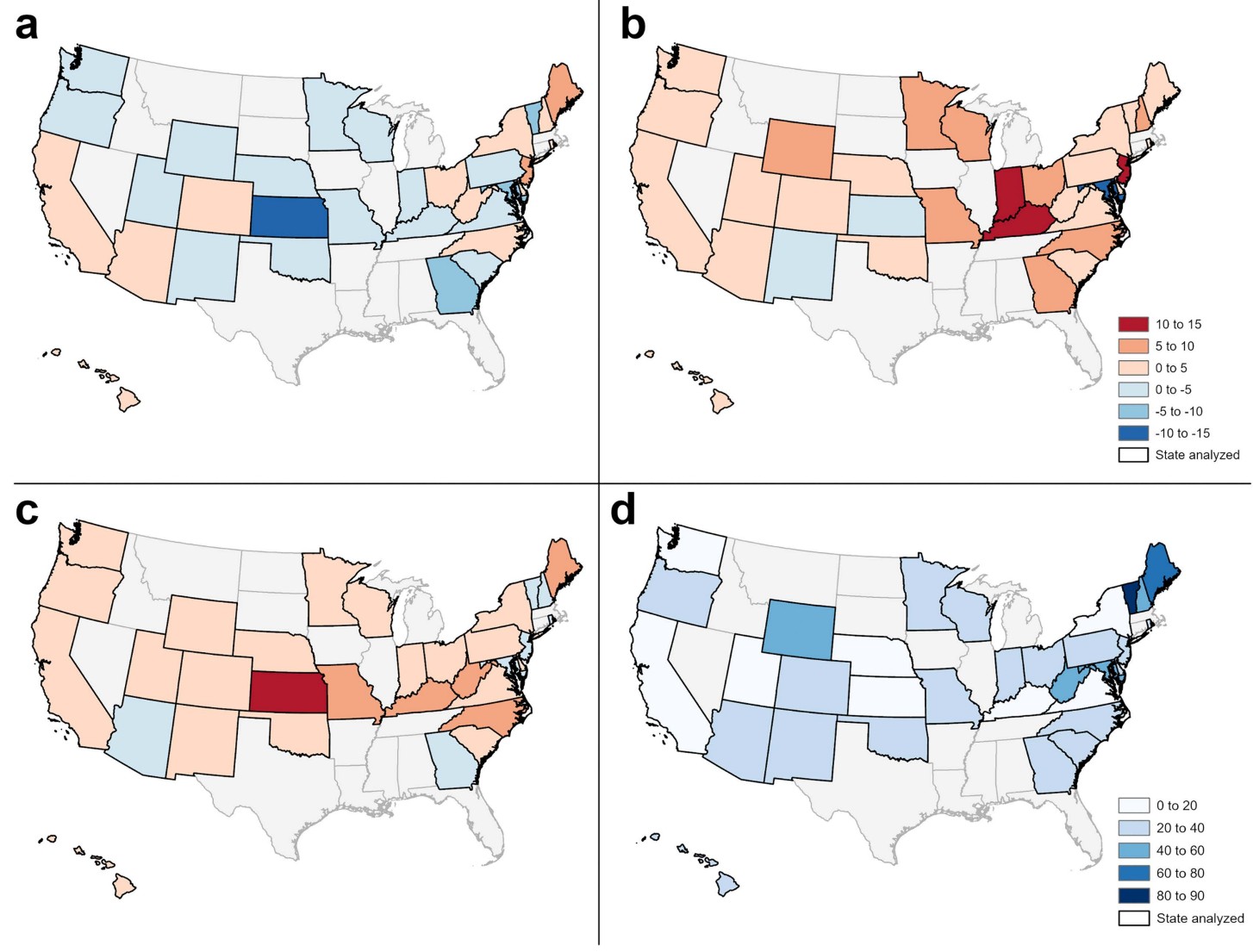

**Fig 2. Differences between statewide monitored sewershed populations and statewide populations.** Maps showing the percentage differences between the aggregated statewide monitored sewershed population proportion (sites sampling August to October 2024) and the statewide population proportion for: **a)** housing type and transportation SVI, **b)** racial and ethnic minority status SVI, **c)** educational attainment, and **d)** median household income. Maps a, b, and c share a common legend. State boundaries are publicly available from https://www.census.gov/geographies/mapping-files/time-series/geo/cartographic-boundary.html.

and lower proportions of people 65 years or older, Whites, and disabled people, and higher proportions of Hispanics, limited English speakers, and those with a bachelor's degree. North Carolina's sewershed population distribution also had significantly different and lower median household income and lower vulnerability related to socioeconomic status, household characteristics, and racial and ethnic minority status than the population distribution in counties (Fig 3). In New York, populations across 188 monitored sewersheds had lower proportions of people with no health insurance, lower median household income and lower housing type and transportation SVI, and racial and ethnic minority status SVI compared to populations in New York's 62 counties. Only vulnerability related to household characteristics was meaningfully shifted towards higher values in New York's sewershed populations (Fig 3). In Colorado, population distributions were similar

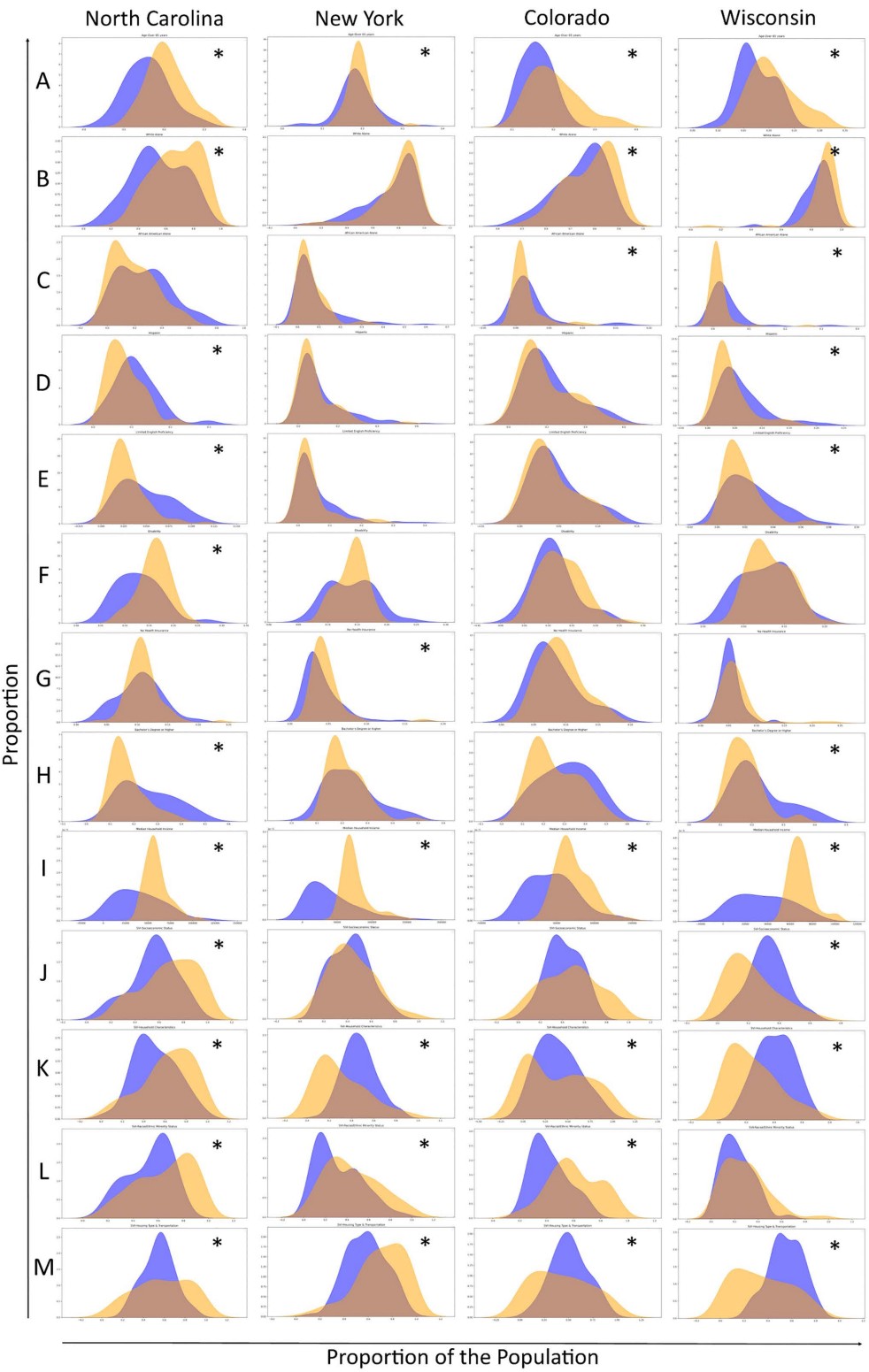

**Fig 3. Distributions of sewershed populations and county populations.** Plots showing the distribution of the proportion of the population for select variables in each sewershed sampling August to October 2024 (orange) and each county (blue) within the state: **A)** 65 years or older, **B)** white, **C)** African American, **D)** Hispanic, **E)** limited English speaking, **F)** disability, **G)** without health insurance, **H)** bachelor's degree or higher, **I)** median household

income, **J)** SES SVI, **K)** household characteristics SVI, **L)** racial and ethnic minority SVI, **M)** housing type and transportation SVI. An asterisk indicates a statistically significant difference (KS statistic p-value <0.05) between the sewershed and county distributions.

between the state's 20 monitored sewersheds and its 64 counties, except for median household income and racial and ethnic minority status vulnerability, where the sewershed populations included lower values (Fig 3). Wisconsin's 47 sewershed populations had lower proportions of people 65 years or older and lower median household income compared to populations in its 72 counties. In Wisconsin, the population distribution in monitored sewersheds was shifted towards higher values for overall SVI as well as vulnerability related to SES, household characteristics, and housing type and transportation (Fig 3). The KS statistic and associated p-value for each variable by state can be found in the supplemental material (S3 Table).

## Comparing characteristics of the aggregated sewershed and statewide populations over time

For the second research question, we explored wastewater-monitored population representativeness over time (January and August, 2021–2024) in four states. In New York, the differences between the wastewater-monitored population and the statewide population have been in the same direction (higher or lower) but have decreased in magnitude as New York increased population coverage by more than tripling the number of monitoring sites between 2022 and 2024. Specifically, the wastewater monitored population in New York has consistently had a higher proportion of African Americans, Asians, and Hispanics and a lower proportion of Whites, and more residents with limited English-speaking ability, and higher social vulnerabilities than the statewide population (Fig 4c). In North Carolina, the monitored sewershed population consistently had a meaningfully different, higher proportion of African Americans, a lower proportion of Whites, and a higher proportion of people with a bachelor's degree, regardless of changes in the number of sampling sites or population coverage. While the direction of differences in North Carolina is consistent over time, the differences decreased when North Carolina had more sites and increased population coverage between January 2023 and January 2024 (Fig 4b). In contrast to findings in New York and North Carolina, Colorado's sewershed population characteristics have been comparable

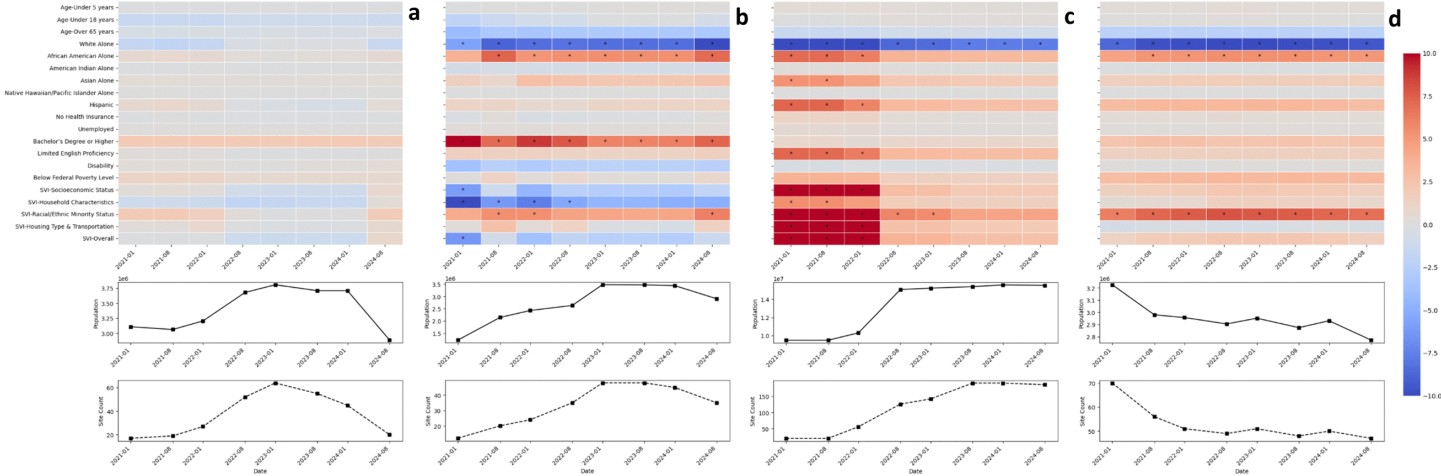

**Fig 4. Differences between statewide monitored sewershed populations and statewide populations and changes in total population and total sites sampled over time.** Heatmaps showing the percentage point differences between the proportion of the population monitored statewide sewersheds and the state and line graphs showing the total sewershed population and number of sewershed sites sampling in January or August of the years 2021, 2022, 2023, and 2024 in **a)** Colorado, **b)** North Carolina, **c)** New York, and **d)** Wisconsin. Dates on the x-axis are aligned to allow for comparisons between plots. An * in the heatmap indicates that the difference was greater than +/-5%.

to the statewide population characteristics over time, even though Colorado's wastewater program expanded to 63 sites in 2023 and then reduced its site count to 20 by August 2024 (Fig 4a). Finally, in Wisconsin, the direction and magnitude of the differences between the sewershed and statewide population characteristics were stable over time, even though the number of monitored sewersheds decreased from 70 to 47 between 2021 and 2024 (Fig 4d).

## Discussion

In this study, we leveraged sewershed polygon data to determine the characteristics of populations included in CDC NWSS and assess whether wastewater-monitored population characteristics are representative of broader populations. Our findings suggest that wastewater monitoring can sometimes capture representative populations, which is useful when tracking widespread respiratory pathogens like SARS-CoV-2 or influenza, while in other cases, a deliberate over-sampling of at-risk groups may be more desirable. This work also highlights the importance of having accurate data about the sewershed population and how under- or over-representation of certain groups can impact the interpretation of wastewater data for public health. In some states, including Colorado, Utah, and South Carolina, wastewater was sampled from sewersheds where the population represented the state population well, indicating that wastewater data can effectively serve as a proxy for disease trends across the state. In other states, we identified differences between the monitored sewershed and the state populations. In general, there was a higher proportion of racial and ethnic minorities and a lower proportion of Whites in sewershed populations. Also, the sewershed population had higher educational attainment, but at the same time, lower median household income and somewhat higher proportions of people living below the federal poverty level compared to the state population. This suggests that although higher education is often linked with higher income, sewershed populations may experience greater income inequity, with a concentration of both high-income earners and low-income earners resulting in a lower median income [21,22]. Factors such as socioeconomic status, income, race, ethnicity, and education, are important determinants of health because they shape individuals' access to resources which then influences their risk for poor health outcomes [23,24]. Including a higher proportion of groups with greater risk factors and a higher burden of disease in wastewater monitoring is beneficial to public health as this can fill critical surveillance gaps and improve public health information about people who may experience a disproportionate burden of disease [25–27]. However, wastewater data should be interpreted in consideration of these similarities and differences because there is a risk that viral levels detected in wastewater could be exaggerated when populations at higher risk of increased disease burden are over-sampled. By recognizing the similarities and differences between sewershed and broader populations, wastewater monitoring data can be effectively integrated with other clinical surveillance sources to inform public health decisions.

When we compared the aggregated sewershed population to the state population, we found that high population coverage did not always correspond with minor differences between the sewershed and the statewide population. Wastewater monitoring programs in Utah, Delaware, Nebraska, Hawaii, and New York included the majority of the state's population, but characteristics of the wastewater-monitored population and the state population differed for some variables. In three states, New York, Delaware, and Hawaii, we observed meaningful differences in race, ethnicity and social vulnerability characteristics even though the wastewater-monitored population included 77%, 63%, and 67% of the state's population, respectively. Sewer connectivity has been linked with certain population demographics and so, in some states, differences between the sewershed and state population characteristics may persist even when a large portion of the population is represented in wastewater monitoring [2]. Conversely, Colorado and South Carolina included a smaller proportion of the state in wastewater monitoring, 28% and 24% respectively, but attributes of the sewershed population closely matched the statewide population across all variables analyzed. These findings suggest that, in these states, although the wastewater-monitored population was relatively small, it reflected the overall diversity of the statewide population.

Since wastewater monitoring largely occurs in communities served by municipal sewer systems in highly populated, urban counties [3], comparing aggregated sewershed population characteristics to statewide population characteristics

may have masked smaller sewersheds' population characteristics. For example, the distribution of the proportion of the population 65 years or older in individual sewersheds was significantly different and lower than the distribution in counties in the four states analyzed even though we did not find meaningful differences between the aggregated sewershed and the state population in 32 states. Understanding whether the elderly are over or under-represented in a state's wastewater data is particularly important to public health because the elderly are at greater risk of severe disease or death due to respiratory illnesses [28]. Additionally, we found that the proportion of the population with a bachelor's degree or higher in sewersheds compared to counties was significantly different only in North Carolina and Wisconsin even though educational attainment was found to be higher in the sewershed population compared to the state population in the majority of states and meaningfully higher in six states. In North Carolina, we found several more differences not observed when comparisons were made at the state level, indicating that the less populous sewershed characteristics differed from populous sewershed characteristics or simply that the ratio of monitored sewersheds compared to counties (35 sewersheds vs 100 counties) necessitated a more granular analysis method. Considering these results and previous research that demonstrated differences between sewershed and county population characteristics [8], states might benefit from assessing population representativeness at different geographic levels. The differences between the distribution of populations in sewersheds and counties also demonstrate the importance of performing epidemiological studies based on the sewershed population rather than the county population when sub-county data are available.

In most states, sampling site composition has fluctuated over time, first expanding to cover a high proportion of the population during the height of the COVID-19 pandemic and then contracting as states scaled back their number of testing sites and added testing for additional pathogens. Increasing the percentage of the state's population included in wastewater monitoring over time did not greatly affect the differences between the wastewater monitored and the statewide population in New York, North Carolina, Wisconsin, and Colorado. Although substantially increasing wastewater monitoring coverage in New York improved wastewater representativeness, differences between the sewershed and state populations remained. Furthermore, changes in the number of sampling sites and population coverage between 2021 and 2024 in North Carolina, Colorado, and Wisconsin had minimal effect on representativeness. This implies that sites which were added or removed were relatively small or had similar population characteristics to the state.

This study was subject to several limitations. First, our analysis may not have included the entire wastewater-monitored population within each state. We assessed sewershed populations using sewershed polygon data submitted to CDC NWSS for ELC-funded sites, which may not include the entirety of wastewater monitoring sites within a state. Also, we included states in our analysis that had submitted a sewershed for at least 80% of their current wastewater monitoring sites or wastewater monitored population, to have sufficient data to estimate wastewater monitored population characteristics while including a majority of states. The analysis could be re-run when complete sewershed geospatial data are available. Second, using a spatial intersect may include people residing outside of the sewershed if a sewershed partially overlaps a block or tract. We attempted to account for this by area-weighting the US Census and ACS population counts and population-weighting the SVI ranks. Third, using a +/-5% difference threshold affected the number of variables deemed meaningfully different. Choosing a lower threshold would reveal more differences in smaller populations, but it also may detect minor variations that lack practical importance in larger populations. Additionally, the KS test flagged differences between sewershed and county distributions as significant for some variables where the proportions of the population were very small, which complicates the interpretation of these results. Finally, the scope of this analysis was limited in several ways. Since sewershed data were only available for wastewater monitoring sites (past and present), we could not assess how characteristics of monitored sewersheds compared to the overall sewered population or how the overall sewered population compared to the state population. Also, we did not attempt to account for the fact that sewershed populations are not static. People who do not reside in a sewershed may travel into a sewershed for school, work, or leisure, or vis-versa, and some sewersheds will experience more population fluctuation than others.

## Future directions

This study provides a framework for assessing wastewater-monitored population characteristics and comparing them with the broader population. Collecting a sewershed polygon for every wastewater sampling site submitting data to NWSS is an ongoing effort, and when complete, this analysis can be repeated to evaluate population representativeness for all states and at the national scale. The geospatial methods described here can be easily adapted to analyze additional population descriptions from US Census or other relevant data or to determine representativeness at different geographic scales, such as state-specific public health regions. Additionally, the method could be expanded to look beyond sewershed population characteristics to include assessing the presence of travel hubs, healthcare facilities, schools, or other points of interest in the sewershed. Future work can incorporate information about sewershed characteristics or representativeness into site selection tools so that wastewater data can efficiently and effectively meet public health needs. For example, when monitoring common respiratory illnesses such as COVID-19 or influenza, wastewater programs could prioritize sampling from sites that closely reflect the broader population to improve the relevance and utility of the data. [29]. Alternatively, they might focus sampling efforts on sewersheds with high proportions of people over 65 years to monitor populations at higher risk for more severe illness, or sewersheds with high numbers of nursing home facilities and hospitals to track hospital-associated infections [30].

## Conclusion

A wastewater sample represents a composite of the people in the sewershed regardless of individuals' demographics, socioeconomic conditions, or access to healthcare. This is the first study to analyze sewershed polygons for wastewater sites participating in CDC NWSS to assess how the wastewater-monitored population compared to broader populations. Through this research, we gained new perspectives about whether certain populations were over- or under-represented in a state's wastewater data and how sewershed population representativeness has changed as states enrolled and dropped sampling sites since NWSS was established in 2021. A better understanding of the sewershed population can provide insight for interpreting pathogens detected in wastewater, thereby strengthening the utility of wastewater monitoring as a tool for protecting the public's health.

## Supporting information

**S1 Fig. Diagram of data analysis workflow.** A diagram showing the data and methods including data collection, geospatial analysis, and calculation of various effect metrics.
(TIF)

**S1 Table. Variables included in the analysis.** Variables are listed with the variable name, source, and geographic level of the data.
(XLSX)

**S2 Table. The percentage point difference between the proportion of the population in the aggregated sewershed (sites sampling August to October 2024) and the proportion of the population in the state.**
(XLSX)

**S3 Table. Results of the two-sample Kolmogorov-Smirnov test.** A KS test compared the distribution of the proportions of the population with certain demographic and social vulnerability characteristics in individual sewersheds and counties within a state for sites sampling August to October 2024. A * denotes that the proportion of the population in group quarters (all types) were very small or zero and so, KS test results should be considered unreliable.
(XLSX)

**S4 Table. The proportion of the population for each variable in the aggregated sewershed (sites sampling August to October 2024) and statewide population.**
(XLSX)

**S5 Table. The ratios between the proportion of the population in the aggregated sewershed (sites sampling August to October 2024) and statewide population for each variable.**
(XLSX)

## Acknowledgments

We thank all state, tribal, local, and territorial health departments, wastewater utilities, and laboratories that have participated in the National Wastewater Surveillance System. Specifically, we thank Arizona Department of Health Services; California Department of Public Health and the many California wastewater utilities participating in the CDC NWSS program; Colorado Department of Public Health and Environment and participating Colorado wastewater utilities; Delaware Public Health; Georgia Department of Public Health; Hawaii State Department of Health, State Laboratories Division, Di, Doris, Yoong, Wen, and Steadmon, Maria; Indiana Department of Health; Kansas Department of Health and Environment; Kentucky Department for Public Health; Maryland Department of the Environment; Maine Department of Health and Human Services; Minnesota Department of Health, Zach Zirnhelt, Epidemiologist Intermediate at MDH Austin Bell, (former) Senior Epidemiologist at MDH; Missouri Department of Health; North Carolina Wastewater Monitoring Network at the North Carolina Department of Health and Human Services; Nebraska Department of Health and Human Services, Derry Stover and Likhitha Duggirala; New Hampshire Department of Health and Human Services; The New Jersey Department of Health's (NJDOH) Communicable Disease Service (CDS) and Public Health Environmental Laboratories (PHEL); New Mexico Department of Health Wastewater Surveillance Program and participating New Mexico municipalities; New York State Department of Health; Ohio Department of Health; Oregon Health Authority Public Health Division and Oregon State University; Pennsylvania Department of Health; Rhode Island Department of Health; South Carolina Department of Public Health (SC DPH); Utah Department of Health and Human Services, Utah Wastewater Surveillance System; Virginia Department of Health; Vermont Department of Health; Washington state Department of Health; Wisconsin Wastewater Monitoring Program, Wisconsin Department of Health Services, Wisconsin State Laboratory of Hygiene, University of Wisconsin-Milwaukee; WV Bureau of Public Health, and WaTCH-WV (Wastewater Testing for Community Health in WV) partners WVU, Marshall University, and participating WWTP operators and WWTP lab personnel; Wyoming Department of Health.

**Disclaimer:** The findings and conclusions in this report are those of the authors and do not necessarily represent the official position of the Centers for Disease Control and Prevention.

## Author contributions

**Conceptualization:** Stacie Reckling, Heather Reese, Helena Mitasova.

**Data curation:** Stacie Reckling.

**Formal analysis:** Stacie Reckling.

**Investigation:** Stacie Reckling, Heather Reese.

**Methodology:** Stacie Reckling, Heather Reese, Lisa P. Oakley, Helena Mitasova.

**Project administration:** Stacie Reckling.

**Software:** Stacie Reckling.

**Supervision:** Lisa P. Oakley, Helena Mitasova.

**Validation:** Stacie Reckling.

**Visualization:** Stacie Reckling, Helena Mitasova.

**Writing – original draft:** Stacie Reckling, Lisa P. Oakley, Helena Mitasova.

**Writing – review & editing:** Stacie Reckling, Heather Reese, Lisa P. Oakley, Helena Mitasova.

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
