## [Decision Letter · Decision Letter 0]

25 Jan 2026

PGPH-D-25-03983

A geospatial analysis comparing wastewater-monitored sewershed and statewide populations for 32 states participating in CDC’s National Wastewater Surveillance System, 2021-2024

Dear Dr. Reckling,

Thank you for submitting your manuscript to PLOS Global Public Health. After careful consideration, we feel that it has merit but does not fully meet PLOS Global Public Health’s publication criteria as it currently stands. Therefore, we invite you to submit a revised version of the manuscript that addresses the points raised during the review process.

We look forward to receiving your revised manuscript.

Kind regards,

Rochelle Holm

Academic Editor

Journal Requirements:

1. We have noticed that you have uploaded Supporting Information files, but you have not included a list of legends. Please add a full list of legends for your Supporting Information files after the references list.

2. We note that your Data Availability Statement is currently as follows: Links to publicly available databases used in this project are provided in the reference section. The data supporting the findings of this study are available in the Supplemental Information. Sewershed polygons are unavailable to access due to concerns about sensitive infrastructure information.

3. Some material included in your submission may be copyrighted. According to PLOS’s copyright policy, authors who use figures or other material (e.g., graphics, clipart, maps) from another author or copyright holder must demonstrate or obtain permission to publish this material under the Creative Commons Attribution 4.0 International (CC BY 4.0) License used by PLOS journals. Please closely review the details of PLOS’s copyright requirements here: PLOS Licenses and Copyright. If you need to request permissions from a copyright holder, you may use PLOS's Copyright Content Permission form.

Potential Copyright Issues:

a. Figure 2: please (a) provide a direct link to the base layer of the map (i.e., the country or region border shape) and ensure this is also included in the figure legend; and (b) provide a link to the terms of use / license information for the base layer image or shapefile. We cannot publish proprietary or copyrighted maps (e.g. Google Maps, Mapquest) and the terms of use for your map base layer must be compatible with our CC-BY 4.0 license.

Additional Editor Comments (if provided):

• Abstract: The abstract would benefit from the inclusion of a few plain-language summary statements, such as: “Currently, NWSS covers X% of the U.S. population. In X of Y states, NWSS provides good representation of the statewide population; however, in Z states, NWSS may over- or underrepresent certain populations based on factors such as race, income, or other demographics.”

• Line 49: Citation #1 is 10 years old. A more appropriate reference would be the 2025 WHO/UNICEF JMP, which includes the proportion of households connected to sewer systems in the United States.

• Lines 63–67: This section is generally not factually accurate. Most publications on WBE over the last five years explicitly include contributing population size in the methods section. This also typically includes population demographics such as income level, race, or ethnicity. These lines should be removed.

• Lines 68–70: In a piped wastewater system in the United States, the number of contributing users is always known, as this information is required under EPA wastewater treatment plant permit requirements. These lines should be removed.

• Table 1 is unclear and needs to be reframed. Using Kentucky as an example: throughout most of the pandemic, there were only five wastewater treatment plants reporting to NWSS (serving a population of less than 1 million). However, Table 1 also includes data through late 2024, when additional ELC-funded sites were added. Reframing the table on a year-by-year basis would better illustrate that some states have experienced decreases in the percentage of population covered, while others (such as Kentucky) have seen increases. As a result, analyses based on late 2024 data differ substantially from those based on 2021 sample sites in terms of population representation. As currently presented, the state population (2020) does not align with the monitored sewershed population (2020) when the 2024 number of sewersheds is applied. For Kentucky, the monitored sewershed population (2020) should be approximately 1,000,000, not 1,548,348 as shown in the table. One option would be to update the table to use 2024 population estimates. This same issue applies to lines 274–301.

Reviewers' comments:

Reviewer's Responses to Questions

**Comments to the Author**

1. Does this manuscript meet PLOS Global Public Health’s publication criteria? Is the manuscript technically sound, and do the data support the conclusions? The manuscript must describe methodologically and ethically rigorous research with conclusions that are appropriately drawn based on the data presented.? Is the manuscript technically sound, and do the data support the conclusions? The manuscript must describe methodologically and ethically rigorous research with conclusions that are appropriately drawn based on the data presented.

Reviewer #1: Yes

Reviewer #2: Yes

2. Has the statistical analysis been performed appropriately and rigorously?

Reviewer #1: No

Reviewer #2: Yes

3. Have the authors made all data underlying the findings in their manuscript fully available (please refer to the Data Availability Statement at the start of the manuscript PDF file)?

The PLOS Data policy requires authors to make all data underlying the findings described in their manuscript fully available without restriction, with rare exception. The data should be provided as part of the manuscript or its supporting information, or deposited to a public repository. For example, in addition to summary statistics, the data points behind means, medians and variance measures should be available. If there are restrictions on publicly sharing data—e.g. participant privacy or use of data from a third party—those must be specified.requires authors to make all data underlying the findings described in their manuscript fully available without restriction, with rare exception. The data should be provided as part of the manuscript or its supporting information, or deposited to a public repository. For example, in addition to summary statistics, the data points behind means, medians and variance measures should be available. If there are restrictions on publicly sharing data—e.g. participant privacy or use of data from a third party—those must be specified.

Reviewer #1: Yes

Reviewer #2: Yes

4. Is the manuscript presented in an intelligible fashion and written in standard English?

Reviewer #1: Yes

Reviewer #2: Yes

Reviewer #1: This manuscript presents a carefully designed and technically sophisticated geospatial analysis of wastewater-monitored populations across multiple spatial and temporal scales. The use of sewershed polygons, dasymetric population estimation, and population-weighted social vulnerability metrics is methodologically sound. However, several aspects of the statistical framework used to quantify and interpret population differences warrant further clarification and, in some cases, reconsideration. These issues primarily concern the choice of effect metrics, the reporting of underlying quantities, and the consistency of inferential approaches.

1. Interpretation of proportional differences based solely on the absolute scale

The analysis of representativeness is based on absolute differences in proportions (percentage point differences), with a fixed ±5% threshold used to identify potentially meaningful discrepancies between aggregated sewershed populations and statewide populations. While absolute differences are intuitive and easy to communicate, their interpretability depends strongly on baseline proportion. For example, an absolute difference of 5 percentage points can correspond to a several-fold relative change for rare characteristics, but only a modest proportional change for highly prevalent characteristics.

From a statistical perspective, relying exclusively on absolute differences may therefore obscure heterogeneity in effect magnitude across variables and lead to differential interpretation depending on scale. This concern is particularly relevant for characteristics with low baseline proportions (e.g., certain racial or group quarters categories), where relative measures such as relative risk (RR) may better capture substantive change. The authors may consider incorporating relative effect measures as complementary summaries, or at a minimum explicitly acknowledge the scale dependence of absolute differences and their implications for interpretation.

2. Reporting of effect metrics without corresponding baseline proportions

Relatedly, the manuscript focuses on reporting differences in proportions without systematically presenting the underlying proportions for the statewide and aggregated sewershed populations. From a statistical reporting standpoint, effect sizes are most interpretable when accompanied by the baseline quantities from which they are derived. Without access to the original proportions, the reader cannot assess the relative magnitude of differences, evaluate potential scale effects, or compute alternative metrics such as relative risk.

Providing the underlying proportions—at least in supplementary materials—would substantially improve transparency and reproducibility, and would allow readers to independently assess the robustness of the conclusions to alternative effect metrics. Including these values would not require reanalysis of the data, but would materially strengthen the interpretability of the results.

3. Consistency of inferential strategy across analytical components

The manuscript employs different inferential strategies across analyses. For comparisons between aggregated sewershed populations and statewide populations, the authors avoid formal hypothesis testing due to non-independence and normality assumptions, opting instead for a heuristic ±5% threshold. In contrast, for n the comparison of distributions between individual sewersheds and counties, the authors apply the Kolmogorov–Smirnov test, a nonparametric test that does not rely on normality assumptions.

While both approaches are defensible in isolation, the rationale for applying a formal nonparametric test in one context but not considering analogous rank-based or distributional methods in the aggregated comparison would benefit from further clarification. Even acknowledging partial dependence between populations, some nonparametric or resampling-based approaches, such as the bootstrapping method, could be explored or at least discussed. Explicitly articulating why such methods were not pursued would improve methodological coherence.

In addition, the choice of ±5% as a cutoff for identifying “potentially meaningful” differences merits further justification. It is unclear whether this threshold is grounded in substantive public health relevance, empirical precedent, or heuristic analogy to conventional statistical significance levels. A brief sensitivity discussion—addressing whether alternative thresholds (e.g., ±1% or ±10%) would materially alter the qualitative conclusions—would help readers assess the robustness of the findings.

Reviewer #2: Overall impressions: This research assesses selected population demographics of wastewater monitoring sewersheds in selected US states in terms of their efficacy in representing the greater populations of which they are a part - either the county or state level. A secondary research question assesses the temporal stability of the relationships between sewershed demographics and state demographics. Through the GIS-based methodology, which apportions sewersheds with demographic data through small area population estimates, the authors elucidate the high variable nature of the relationships between sewershed demographics and those of the larger spatial units they are nested within. The compelling argument that this paper makes to me is that as a general practice, based on the current distribution of NWSS monitoring sites we should not be using wastewater monitoring data to make claims about public health situations in areas beyond the the boundaries of the sewersheds they are sampling. This is very important, as data such as wastewater monitoring data may be used to draw conclusions that those data do not support. The manuscript would be strengthened by addressing this more directly and citing instances where this fallacy occurs. The research also supports the notion that sewersheds can be designed to capture specific populations that may be more or less representative of larger spatial units. Understanding the population demographics or a prospective sewershed prior to monitoring could yield better outcomes and lower costs.

General Feedback: The paper is well organized and provides good background and context on wastewater monitoring and the NWSS from which the spatial data were drawn. I recommend revising the methods section for clarity, as it is quite dense and laborious right now. I think some tables and figures could greatly aid the reader in understanding the data and processing workflows, which could then facilitate condensing the body text. I have provided specific comments in the marked up PDF file.

On the quantitative side, the use of the arbitrary “meaningful difference” threshold of +/-5% is of concern. If there are existing usages of this threshold in the literature it would be helpful to cite them. Perhaps there are other ways to evaluate the significance of these differences?

The figures are generally good, but please see my comment regarding the symbology on the maps. I recommend considering alternative data classifications with fewer classes.

**Do you want your identity to be public for this peer review?** For information about this choice, including consent withdrawal, please see our Privacy Policy..

Reviewer #1: No

Reviewer #2: No

---

## [Editor Report · Decision Letter 1]

19 Mar 2026

A geospatial analysis comparing wastewater-monitored sewershed and statewide populations for 32 states participating in CDC’s National Wastewater Surveillance System, 2021-2024

PGPH-D-25-03983R1

Dear GIS scientist Reckling,

We are pleased to inform you that your manuscript 'A geospatial analysis comparing wastewater-monitored sewershed and statewide populations for 32 states participating in CDC’s National Wastewater Surveillance System, 2021-2024' has been provisionally accepted for publication in PLOS Global Public Health.

Best regards,

Rochelle Holm

Academic Editor

The authors have done an excellent job revising their manuscript.